# Domain Generalization via Model-Agnostic Learning of Semantic Features

**Qi Dou    Daniel C. Castro    Konstantinos Kamnitsas    Ben Glocker**
Biomedical Image Analysis Group, Imperial College London, UK
{qi.dou,dc315,kk2412,b.glocker}@imperial.ac.uk

## Abstract

Generalization capability to unseen domains is crucial for machine learning models when deploying to real-world conditions. We investigate the challenging problem of domain generalization, i.e., training a model on multi-domain source data such that it can directly generalize to target domains with unknown statistics. We adopt a model-agnostic learning paradigm with gradient-based meta-train and meta-test procedures to expose the optimization to domain shift. Further, we introduce two complementary losses which explicitly regularize the semantic structure of the feature space. Globally, we align a derived soft confusion matrix to preserve general knowledge about inter-class relationships. Locally, we promote domain-independent class-specific cohesion and separation of sample features with a metric-learning component. The effectiveness of our method is demonstrated with new state-of-the-art results on two common object recognition benchmarks. Our method also shows consistent improvement on a medical image segmentation task.

## 1    Introduction

Machine learning methods have achieved remarkable success, under the assumption that training and test data are sampled from the same distribution. In real-world applications, this assumption is often violated as conditions for data acquisition may change, and a trained system may fail to produce accurate predictions for unseen data with domain shift. To tackle this issue, domain adaptation algorithms normally learn to align source and target data in a domain-invariant discriminative feature space [6, 11, 19, 32, 33, 42, 43, 50, 51]. These methods rely on access to a few labelled [6, 42, 50] or unlabelled [11, 19, 32, 33, 43, 51] data samples from the target distribution during training.

An arguably harder problem is domain generalization, which aims to train a model using multi-domain source data, such that it can directly generalize to new domains without need of retraining. This setting is very different to domain adaptation as no information about the new domains is available, a scenario that is encountered in real-world applications. In the field of healthcare, for example, medical images acquired at different sites can differ significantly in their data distribution, due to varying scanners, imaging protocols or patient cohorts. At deployment, each new hospital can be regarded as a new domain but it is impractical to collect data each time to adapt a trained system. Learning a model which directly generalizes to new clinical sites would be of great practical value.

Domain generalization is an active research area with a number of approaches being proposed. As no *a priori* knowledge of the target distribution is available, the key question is how to guide the model learning to capture information which is discriminative for the specific task but insensitive to changes of domain-specific statistics. For computer vision applications, the aim is to capture general semantic features for object recognition. Previous work has demonstrated that this can be investigated through regularization of the feature space, e.g., by minimizing divergence between marginal distributions of data sources [35], or joint consideration of the class conditional distributions [30]. Li et al. [28] use adversarial feature alignment via maximum mean discrepancy. Leveraging distance metrics of feature

vectors is another method [20, 34]. Model-agnostic meta-learning [10] is a recent gradient-based method for fast adaptation of models to new conditions, e.g., a new task at few-shot learning. Meta-learning has been introduced to address domain generalization [1, 26, 31], by adopting an episodic training paradigm, i.e., splitting the available source domains into meta-train and meta-test at each iteration, to simulate domain shift. Promising performance has been demonstrated by deriving the loss from a task error [26], a classifier regularizer [1], or a predictive feature-critic module [31].

We introduce two complementary losses which explicitly regularize the semantic structure of the feature space via a model-agnostic episodic learning procedure. Our optimization objective encourages the model to learn semantically consistent features across training domains that may generalize better to unseen domains. Globally, we align a derived soft confusion matrix to preserve inter-class relationships. Locally, we use a metric-learning component to encourage domain-independent while class-specific cohesion and separation of sample features. The effectiveness of our approach is demonstrated with new state-of-the-art performance on two common object recognition benchmarks. Our method also shows consistent improvement on a medical image segmentation task. Code for our proposed method is available at: https://github.com/biomedia-mira/masf.

## 2 Related Work

**Domain adaptation** is based on the central theme of bounding the target error by the source error plus a discrepancy metric between the target and the source [2]. This is practically performed by narrowing the domain shift between the target and source either in input space [19], feature space [6, 11, 32, 42, 51], or output space [33, 43, 49], generally using maximum mean discrepancy [15, 46] or adversarial learning [14]. The success of methods operating on feature representations motivates us to optimize the semantic feature space for domain generalization in this paper.

**Domain generalization** aims to generalize models to unseen domains without knowledge about the target distribution during training. Different methods have been proposed for learning generalizable and transferable representations. A promising direction is to extract task-specific but domain-invariant features [12, 28, 30, 34, 35]. Muandet et al. [35] propose a domain-invariant component analysis method with a kernel-based optimization algorithm to minimize the dissimilarity across domains. Ghifary et al. [12] learn multi-task auto-encoders to extract invariant features which are robust to domain variations. Li et al. [30] consider the conditional distribution of label space over input space, and minimize discrepancy of a joint distribution. Motiian et al. [34] use contrastive loss to guide samples from the same class being embedded nearby in latent space across data sources. Li et al. [28] extend adversarial autoencoders by imposing maximum mean discrepancy measure to align multi-domain distributions. Instead of harmonizing the feature space, others use low-rank parameterized CNNs [25] or decompose network parameters to domain-specific/-invariant components [22]. Data augmentation strategies, such as gradient-based domain perturbation [47] or adversarially perturbed samples [53] demonstrate effectiveness for model generalization. A recent method with state-of-the-art performance is JiGen [3], which leverages self-supervised signals by solving jigsaw puzzles.

**Meta-learning** (a.k.a. learning to learn [44, 48]) is a long standing topic exploring the training of a meta-learner that learns how to train particular models [10, 29, 36, 37]. Recently, gradient-based meta-learning methods [10, 36] have been successfully applied to few-shot learning, with a procedure purely leveraging gradient descent. The episodic training paradigm, originated from model-agnostic meta-learning (MAML) [10], has been introduced to address domain generalization [1, 26, 27, 31]. Epi-FCR [27] alternates domain-specific feature extractors and classifiers across domains via episodic training, but without using inner gradient descent update. The method of MLDG [26] closely follows the update rule of MAML, back-propagating the gradients from an ordinary task loss on meta-test data. A limitation is that using the task objective might be sub-optimal, as it is highly abstracted from the feature representations (only using class probabilities). Moreover, it may not well fit the scenario where target data are unavailable (as pointed out by Balaji et al. [1]). A recent method, MetaReg [1], learns a regularization function (e.g., weighted $L_1$ loss) particularly for the network's classification layer, excluding the feature extractor. Instead, Li et al. [31] propose a feature-critic network which learns an auxiliary meta loss (producing a non-negative scalar) depending on output of the feature extractor. Both [1] and [31] lack notable guidance from semantics of feature space, which may contain crucial domain-independent 'general knowledge' for model generalization. Our method is orthogonal to previous work, proposing to enforce semantic features via global class alignment and local sample clustering, with losses explicitly derived in an episodic learning procedure.

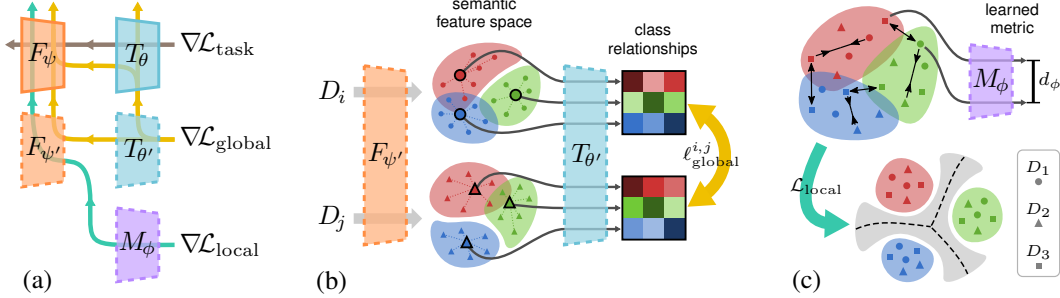

Figure 1: An overview of the proposed model-agnostic learning of semantic features (MASF): (a) episodic training under simulated domain shift, with gradient flows indicated; (b) global alignment of class relationships; (c) local sample clustering, towards cohesion and separation. $F_\psi$ and $T_\theta$ are the feature extractor and the task net, $F_{\psi'}$ and $T_{\theta'}$ are their updated versions by inner gradient descent on the task loss $\mathcal{L}_{\text{task}}$, the $M_\phi$ is a metric embedding net, and $D_k$ denotes different source domains.

## 3  Method

In the following, we denote input and label spaces by $\mathcal{X}$ and $\mathcal{Y}$, the domains $\mathcal{D} = \{D_1, D_2, \ldots, D_K\}$ are different distributions on the joint space $\mathcal{X} \times \mathcal{Y}$. Since domain generalization involves a common predictive task, the label space is shared by all domains. In each domain, samples are drawn from a dataset $D_k = \{(\mathbf{x}_n^{(k)}, y_n^{(k)})\}_{n=1}^{N_k}$ where $N_k$ is the number of labeled data points in the $k$-th domain. The domain generalization (DG) setting further assumes the existence of domain-invariant patterns in the inputs (e.g. *semantic features*), which can be extracted to learn a label predictor that performs well across seen and unseen domains. Unlike domain adaptation, DG assumes no access to observations from or explicit knowledge about the target distribution.

In this work, we consider a classification model composed of a feature extractor, $F_\psi : \mathcal{X} \to \mathcal{Z}$, where $\mathcal{Z}$ is a feature space (typically much lower-dimensional than $\mathcal{X}$), and a task network, $T_\theta : \mathcal{Z} \to \mathbb{R}^C$, where $C$ is the number of classes in $\mathcal{Y}$. The final class predictions are given by $p(y \mid \mathbf{x}; \psi, \theta) = \hat{\mathbf{y}} = \text{softmax}(T_\theta(F_\psi(\mathbf{x})))$, where $\text{softmax}(\mathbf{a}) = e^{\mathbf{a}} / \sum_r e^{a_r}$.[1] The parameters $(\psi, \theta)$ are optimized with respect to a task-specific loss $\mathcal{L}_{\text{task}}$, e.g. cross-entropy: $\ell_{\text{task}}(y, \hat{\mathbf{y}}) = -\sum_c \mathbf{1}[y = c] \log \hat{y}_c$.

Although the minimization of $\mathcal{L}_{\text{task}}$ may produce highly discriminative features $\mathbf{z} = F_\psi(\mathbf{x})$, and hence an excellent predictor for data from the training domains, nothing in this process prevents the model from overfitting to the source domains and suffering from degradation on unseen test domains. We therefore propose to optimize the feature space such that its semantic structure is insensitive to different training domains, and generalize better to new unseen domains. Figure 1 gives an overview of our **m**odel-**a**gnostic learning of **s**emantic **f**eatures (MASF), which we will detail in this section.

### 3.1  Model-Agnostic Learning with Episodic Training

The key of our learning procedure is an episodic training scheme, originated from model-agnostic meta-learning [10], to expose the model optimization to distribution mismatch. In line with our goal of domain generalization, the model is trained on a sequence of simulated *episodes* with domain shift. Specifically, at each iteration, the available domains $\mathcal{D}$ are randomly split into sets of meta-train $\mathcal{D}_{\text{tr}}$ and meta-test $\mathcal{D}_{\text{te}}$ domains. The model is trained to semantically perform well on held-out $\mathcal{D}_{\text{te}}$ after being optimized with one or more steps of gradient descent with $\mathcal{D}_{\text{tr}}$ domains. In our case, the feature extractor's and task network's parameters, $\psi$ and $\theta$, are first updated from the task-specific supervised loss $\mathcal{L}_{\text{task}}$ (e.g. cross-entropy for classification), computed on meta-train:

$$(\psi', \theta') = (\psi, \theta) - \alpha \nabla_{\psi, \theta} \mathcal{L}_{\text{task}}(\mathcal{D}_{\text{tr}}; \psi, \theta), \tag{1}$$

where $\alpha$ is a learning-rate hyperparameter. This results in a predictive model $T_{\theta'} \circ F_{\psi'}$ with improved task accuracy on the meta-train source domains, $\mathcal{D}_{\text{tr}}$.

Once this optimized set of parameters has been obtained, we can apply a meta-learning step, aiming to enforce certain properties that we desire the model to exhibit on held-out domain $\mathcal{D}_{\text{te}}$. Crucially, the

**Algorithm 1** Model-agnostic learning of semantic features for domain generalization

---

**Input:** Source training domains $\mathcal{D} = \{D_k\}_{k=1}^{K}$; hyperparameters $\alpha, \eta, \gamma, \beta_1, \beta_2 > 0$
**Output:** Feature extractor $F_\psi$, task network $T_\theta$, embedding network $M_\phi$

1: **repeat**
2:      Randomly split source domains $\mathcal{D}$ into disjoint meta-train $\mathcal{D}_{\text{tr}}$ and meta-test $\mathcal{D}_{\text{te}}$
3:      $(\psi', \theta') \leftarrow (\psi, \theta) - \alpha \nabla_{\psi, \theta} \mathcal{L}_{\text{task}}(\mathcal{D}_{\text{tr}}; \psi, \theta)$
4:      Compute global class alignment loss:
       $\mathcal{L}_{\text{global}} \leftarrow \frac{1}{|\mathcal{D}_{\text{tr}}|} \sum_{D_i \in \mathcal{D}_{\text{tr}}} \frac{1}{|\mathcal{D}_{\text{te}}|} \sum_{D_j \in \mathcal{D}_{\text{te}}} \ell_{\text{global}}(D_i, D_j; \psi', \theta')$           // Section 3.2
5:      Compute local sample clustering loss:
       $\mathcal{L}_{\text{local}}(\mathcal{D}; \psi', \phi) \leftarrow \mathbb{E}_\mathcal{D}[\ell_{\text{con}}^{n,m}] \ \text{ or } \ \mathbb{E}_\mathcal{D}[\ell_{\text{tri}}^{a,p,n}]$                    // Section 3.3
6:      $\mathcal{L}_{\text{meta}} \leftarrow \beta_1 \mathcal{L}_{\text{global}} + \beta_2 \mathcal{L}_{\text{local}}$
7:      $(\psi, \theta) \leftarrow (\psi, \theta) - \eta \nabla_{\psi, \theta} (\mathcal{L}_{\text{task}} + \mathcal{L}_{\text{meta}})$
8:      $\phi \leftarrow \phi - \gamma \nabla_\phi \mathcal{L}_{\text{local}}$
9: **until** convergence

---

objective function quantifying these properties, $\mathcal{L}_{\text{meta}}$, is computed based on the updated parameters, $(\psi', \theta')$, and the gradients are computed towards the original parameters, $(\psi, \theta)$. Intuitively, besides the task itself, the training procedure is learning how to generalize under domain shift. In other words, parameters are updated such that future updates with given source domains also improve the model regarding some generalizable aspects on unseen target domains.

In particular, we desire the feature space to encode *semantically relevant* properties: features from different domains should respect inter-class relationships, and they should be compactly clustered by class labels regardless of domains (cf. Alg. 1). In the remainder of this section we describe the design of our semantic meta-objective, $\mathcal{L}_{\text{meta}} = \beta_1 \mathcal{L}_{\text{global}} + \beta_2 \mathcal{L}_{\text{local}}$, composed of a *global* class alignment term and a *local* sample clustering term, with weighting coefficients $\beta_1, \beta_2 > 0$.

### 3.2 Global Class Alignment Objective

Relationships between class concepts exist in purely semantic space, independent of changes in the observation domain. In light of this, compared with individual hard label prediction, aligning class relationships can promote more transferable knowledge towards model generalization. This is also noted by Tzeng et al. [50] in the context of domain adaptation, by aggregating the output probability distribution when fine-tuning the model on a few labelled target data. In contrast to their work, our goal is to structure the feature space itself to preserve learned class relationships on unseen data, by means of explicit regularization.

Specifically, we formulate this objective in a manner that imposes a *global* layout of extracted features, such that the relative locations of features from different classes embody the inherent similarity in semantic structures. Inspired by knowledge distillation from neural networks [18], we exploit what the model has learned about class ambiguities—in the form of per-class soft labels—and enforce them to be consistent between $\mathcal{D}_{\text{tr}}$ and $\mathcal{D}_{\text{te}}$ domains. For each domain $k$, we summarize the model's current 'concept' of each class $c$ by computing the class-specific mean feature vectors $\bar{\mathbf{z}}_c^{(k)}$:

$$\bar{\mathbf{z}}_c^{(k)} = \frac{1}{N_k^{(c)}} \sum_{n: y_n^{(k)} = c} F_{\psi'}(\mathbf{x}_n^{(k)}) \approx \mathbb{E}_{D_k}[F_{\psi'}(\mathbf{x}) \,|\, y = c]\,, \tag{2}$$

where $N_k^{(c)}$ is the number of samples in domain $\mathcal{D}_k$ labelled as class $c$. The obtained $\bar{\mathbf{z}}_c^{(k)}$ conveys how samples from a particular class are generally represented. It is then forwarded to the task network $T_{\theta'}$, for computing soft label distributions $\mathbf{s}_c^{(k)}$ with a 'softened' softmax at temperature $\tau > 1$ [18]:

$$\mathbf{s}_c^{(k)} = \text{softmax}(T_{\theta'}(\bar{\mathbf{z}}_c^{(k)})/\tau)\,. \tag{3}$$

The collection of soft labels $[\mathbf{s}_c^{(k)}]_{c=1}^{C}$ represents a kind of 'soft confusion matrix' associated with a particular domain, encoding the inter-class relationships learned by the model. Such relationships should be preserved as general semantics on meta-test after updating the classification model on meta-train (e.g., cartoon dogs are more easily misclassified as horses than as houses, which likely holds in unseen domains). Standard supervised training with $\mathcal{L}_{\text{task}}$ focuses only on the dominant hard

label prediction, there is no reason *a priori* for consistency of such inter-class alignment. We therefore propose to align the soft class confusion matrix between two domains $D_i \in \mathcal{D}_{\mathrm{tr}}$ and $D_j \in \mathcal{D}_{\mathrm{te}}$, by minimising their symmetrized Kullback–Leibler (KL) divergence, averaged over all $C$ classes:

$$\ell_{\mathrm{global}}(D_i, D_j; \psi', \theta') = \frac{1}{C} \sum_{c=1}^{C} \frac{1}{2} [D_{\mathrm{KL}}(\mathbf{s}_c^{(i)} \,\|\, \mathbf{s}_c^{(j)}) + D_{\mathrm{KL}}(\mathbf{s}_c^{(j)} \,\|\, \mathbf{s}_c^{(i)})] , \qquad (4)$$

where $D_{\mathrm{KL}}(\mathbf{p} \,\|\, \mathbf{q}) = \sum_r p_r \log \frac{p_r}{q_r}$. Other symmetric divergences such as Jensen–Shannon (JS) could also be considered, although our preliminary experiments showed no significant difference with JS over symm. KL. Finally, the global class alignment loss, $\mathcal{L}_{\mathrm{global}}(\mathcal{D}_{\mathrm{tr}}, \mathcal{D}_{\mathrm{te}}; \psi', \theta')$, is calculated as the average of $\ell_{\mathrm{global}}(D_i, D_j; \psi', \theta')$ over all pairs of available meta-train and meta-test domains, $(D_i, D_j) \in \mathcal{D}_{\mathrm{tr}} \times \mathcal{D}_{\mathrm{te}}$ (cf. Alg. 1). The complexity of this computation is not problematic in practice, since the number of domains selected in a training mini-batch is limited (as with the form in MAML [10]), and in our experiments we took $|\mathcal{D}_{\mathrm{tr}}| = 2$ and $|\mathcal{D}_{\mathrm{te}}| = 1$.

### 3.3  Local Sample Clustering Objective

In addition to promoting the alignment of class relationships across domains with $\mathcal{L}_{\mathrm{global}}$ as defined above, we further encourage robust semantic features that *locally* cluster according to class regardless of the domain. This is crucial, as neither of the class-prediction-based losses ($\mathcal{L}_{\mathrm{task}}$ or $\mathcal{L}_{\mathrm{global}}$) ensure that features of samples in the same class will lie close to each other and away from those of different classes, a.k.a. feature compactness [21]. If the model cannot project the inputs to the semantic feature clusters with domain-independent class-specific cohesion and separation, the predictions may suffer from ambiguous decision boundaries, and still be sensitive to unseen kinds of domain shift. We therefore propose a local regularization objective $\mathcal{L}_{\mathrm{local}}$ to boost robustness, by increasing the compactness of class-specific clusters while reducing their overlap. Note how this is complementary to the global class alignment of semantically structuring the relative locations among class clusters.

Our preliminary experiments revealed that applying such regularization explicitly onto the features may constrain the optimization for $\mathcal{L}_{\mathrm{task}}$ and $\mathcal{L}_{\mathrm{global}}$ too heavily, hurting generalization performance on unseen domain. We thus take a *metric-learning* approach, introducing an embedding network $M_\phi$ that operates on the extracted features, $\mathbf{z} = F_{\psi'}(\mathbf{x})$. This component represents a learnable distance function [5] between feature vectors (rather than between raw inputs):

$$d_\phi(\mathbf{z}_n, \mathbf{z}_m) = \|\mathbf{e}_n - \mathbf{e}_m\|_2 = \|M_\phi(\mathbf{z}_n) - M_\phi(\mathbf{z}_m)\|_2 . \qquad (5)$$

The sample pairs $(n, m)$ are randomly drawn from all source domains $\mathcal{D}$, because we expect the updated $F_{\psi'}$ will harmonize the semantic feature space of $\mathcal{D}_{\mathrm{te}}$ with that of $\mathcal{D}_{\mathrm{tr}}$, in terms of class-specific clustering regardless of domains. The computed embeddings, $\mathbf{e} = M_\phi(\mathbf{z})$, can then be optimized with any suitable metric-learning loss $\mathcal{L}_{\mathrm{local}}(\mathcal{D}; \psi', \phi)$ to regularize the local sample clustering. Under mild domain shift, the contrastive loss [16] is a sensible choice, as it attempts to separately collapse each group of same-class exemplars to a distinct single point. It might however be over-restrictive for more extreme situations, wherein domains are related rather semantically, but with wildly distinct low-level statistics. For such cases, we propose instead to employ the triplet loss [45].

**Contrastive loss** is computed for pairs of samples, attracting samples of the same class and repelling samples of different classes [16]. Instead of pushing clusters apart to infinity, the repulsion range is bounded by a distance margin $\xi$. Our contrastive loss for a pair of samples $(n, m)$ is defined as:

$$\ell_{\mathrm{con}}^{n,m} = \begin{cases} d_\phi(\mathbf{z}_n, \mathbf{z}_m)^2 , & \text{if } y_n = y_m \\ (\max\{0, \ \xi - d_\phi(\mathbf{z}_n, \mathbf{z}_m)\})^2 , & \text{if } y_n \neq y_m \end{cases} . \qquad (6)$$

The total loss for a training mini-batch, $\mathcal{L}_{\mathrm{local}}$, is normally averaged over all pairs of samples. In cases where full $O(N^2)$ enumeration is intractable—e.g. image segmentation, which would involve all pairs of pixels in all images—we can obtain an unbiased $O(N)$ estimator of the loss by e.g. shuffling the samples and iterating over $(2i - 1, 2i)$ pairs with $i = 1, \ldots, \lfloor N/2 \rfloor$.

**Triplet loss** aims to make pairs of samples from the same class closer than pairs from different classes, by a certain margin $\xi$ [45]. Given one 'anchor' sample $a$, one 'positive' sample $p$ (with $y_a = y_p$), and one 'negative' sample $n$ (with $y_a \neq y_n$), we compute their triplet loss as follows:

$$\ell_{\mathrm{tri}}^{a,p,n} = \max\{0, \ d_\phi(\mathbf{z}_a, \mathbf{z}_p)^2 - d_\phi(\mathbf{z}_a, \mathbf{z}_n)^2 + \xi\} . \qquad (7)$$

Schroff et al. [45] argue that judicious triplet selection is essential for good convergence, as many triplets may already satisfy this constraint and others may be too hard to contribute meaningfully to the learning process. Here we adopt their proposed online 'semi-hard' triplet mining strategy, and $\mathcal{L}_{\mathrm{local}}$ is the average over all selected triplet pairs.

## 4 Experiments

We evaluate and compare our method on three datasets: 1) the classic VLCS domain generalization benchmark for image classification, 2) the recently introduced PACS benchmark for object recognition with challenging domain shift, 3) a real-world medical imaging task of tissue segmentation in brain MRI. Results with an in-depth analysis and ablation study are presented in the following.

### 4.1 VLCS Dataset

VLCS [8] is a classic benchmark for domain generalization, which includes images from four datasets: PASCAL VOC2007 (V) [7], LabelMe (L) [41], Caltech (C) [9], and SUN09 (S) [4]. The multi-class object recognition task includes five classes: bird, car, chair, dog and person. We follow previous work [3, 27, 34] of using the publicly available pre-extracted DeCAF$_6$ features (4096-dimensional vector) for leave-one-domain-out validation with randomly dividing each domain into $70\%$ training and $30\%$ test, inputting to two fully connected layers with output size of 1024 and 128 with ReLU activation. For our metric embedding $M_\phi$ (inputting the 128-dimensional vector), we use two fully connected layers with output size of 128 and 64. The triplet loss is adopted for computing $\mathcal{L}_{\mathrm{local}}$, with coefficient $\beta_2 = 0.005$, such that it is in a similar scale to $\mathcal{L}_{\mathrm{task}}$ and $\mathcal{L}_{\mathrm{global}}$ ($\beta_1 = 1$). We use the Adam optimizer [23] with $\eta$ initialized to $10^{-3}$ and exponentially decayed by $2\%$ every $1k$ iterations. For the inner optimization to obtain $(\psi', \theta')$, we clip the gradients by norm (threshold by 2.0) to prevent them from exploding, since this step uses plain, non-adaptive gradient descent (with learning rate $\alpha = 10^{-5}$). Note that, although performing gradient descent on $\mathcal{L}_{\mathrm{meta}}$ involves second-order gradients on $(\psi, \theta)$, their computation does not incur a substantial overhead in training time [10]. We also employ an Adam optimizer for the meta-updates of $\phi$ with learning rate $\gamma = 10^{-5}$ without decay. The batch size is 128 for each source domain, with an Nvidia TITAN Xp 12 GB GPU. The metric-learning margin hyperparameter $\xi$ was chosen heuristically based on observing the distances within and between the clusters of class features. For our results, we report the average and standard deviation over three independent runs.

**Results.** Table 1 shows the object recognition accuracies on different target domains. Our DeepAll baseline—i.e., merging all source domains and training $F_\psi \circ T_\theta$ by standard supervised learning on $\mathcal{L}_{\mathrm{task}}$ with the same hyperparameters—achieves an average accuracy of $72.19\%$ over four domains. Using our episodic training paradigm with regularizations on semantic feature space, we improve the performance to $74.11\%$, setting the state-of-the-art accuracy on VLCS. We compare with eight different methods (cf. Section 2) which report previous best results on this benchmark. CCSA [34] combines contrastive loss together with ordinary cross-entropy without using episodic meta-update paradigm. Notably, our approach outperforms MLDG [26], indicating that explicitly encouraging semantic properties in the feature space is superior to using a highly-abstracted task loss on meta-test.

Table 1: Domain generalization results on VLCS dataset with object recognition accuracy (%).

| Source | Target | D-MTAE [12] | CIDDG [30] | CCSA [34] | DBADG [25] | MMD-AAE [28] | MLDG [26] | Epi-FCR [27] | JiGen [3] | DeepAll (Baseline) | MASF (Ours) |
|---|---|---|---|---|---|---|---|---|---|---|---|
| L,C,S | V | 63.90 | 64.38 | 67.10 | 69.99 | 67.70 | 67.7 | 67.1 | 70.62 | 68.67±0.09 | 69.14±0.19 |
| V,C,S | L | 60.13 | 63.06 | 62.10 | 63.49 | 62.60 | 61.3 | 64.3 | 60.90 | 63.10±0.11 | 64.90±0.08 |
| V,L,S | C | 89.05 | 88.83 | 92.30 | 93.63 | 94.40 | 94.4 | 94.1 | 96.93 | 92.86±0.13 | 94.78±0.16 |
| V,L,C | S | 61.33 | 62.10 | 59.10 | 61.32 | 64.40 | 65.9 | 65.9 | 64.30 | 64.11±0.17 | 67.64±0.12 |
| Average | | 68.60 | 69.59 | 70.15 | 72.11 | 72.28 | 72.3 | 72.9 | 73.19 | 72.19 | 74.11 |

### 4.2 PACS Dataset

The PACS dataset [25] is a recent benchmark with more severe distribution shift between domains, making it more challenging than VLCS. It consists of four domains: art painting, cartoon, photo, sketch, with objects from seven classes: dog, elephant, giraffe, guitar, house, horse, person. Following practice in the literature [1, 3, 26, 27], we also use leave-one-domain-out cross-validation, i.e., training

Table 2: Domain generalization results on PACS dataset with recognition accuracy (%) using AlexNet.

| Source | Target | D-MTAE [12] | CIDDG [30] | DBADG [25] | MLDG [26] | Epi-FCR [27] | MetaReg [1] | JiGen [3] | DeepAll (Baseline) | MASF (Ours) |
|--------|--------|-------------|------------|------------|-----------|--------------|-------------|-----------|--------------------|-------------|
| C,P,S | Art painting | 60.27 | 62.70 | 62.86 | 66.23 | 64.7 | 69.82 | 67.63 | 67.60±0.21 | 70.35±0.33 |
| A,P,S | Cartoon | 58.65 | 69.73 | 66.97 | 66.88 | 72.3 | 70.35 | 71.71 | 68.87±0.22 | 72.46±0.19 |
| A,C,S | Photo | 91.12 | 78.65 | 89.50 | 88.00 | 86.1 | 91.07 | 89.00 | 89.20±0.24 | 90.68±0.12 |
| A,C,P | Sketch | 47.68 | 64.45 | 57.51 | 58.96 | 65.0 | 59.26 | 65.18 | 61.13±0.30 | 67.33±0.12 |
| | Average | 64.48 | 68.88 | 69.21 | 70.01 | 72.0 | 72.62 | 73.38 | 71.70 | 75.21 |

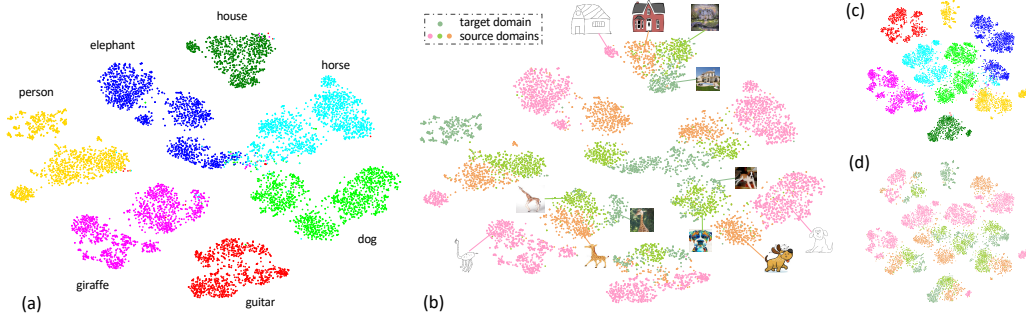

Figure 2: The t-SNE visualization of extracted features from $F_\psi$, using our proposed (a-b) MASF and the (c-d) DeepAll model on PACS dataset. In (a) and (c), the different colors indicate different classes; correspondingly in (b) and (d), the different colors indicate different domains.

on three domains and testing on the remaining unseen one, and adopt an AlexNet [24] pre-trained on ImageNet [40]. The metric embedding $M_\phi$ is connected to the last fully connected layer (i.e., fc7 layer with a 4096-dimesional vector), by stacking two fully connected layers with output size of 1024 and 256. For the $\mathcal{L}_{\text{local}}$, we also use the triplet loss with $\beta_2 = 0.005, \beta_1 = 1.0$, particularly considering the severe domain shift. We initialize learning rates $\alpha = \eta = \gamma = 10^{-5}$ and clip inner gradients by norm. The batch size is 128 for each source domain.

**Results.** Table 2 summarizes the results of object recognition on PACS dataset with a comparison to previous work (noting that not all compared methods reported results on both VLCS and PACS). MLDG [26] and MetaReg [1] employ episodic training with meta-learning, but from different angles in terms of the meta learner's objective (Li et al. [26] minimize task error, Balaji et al. [1] learn a classifier regularizer). The promising results for [1, 26, 27] indicate that exposing the training procedure to domain shift benefits model generalization to unseen domains. Our method further explicitly considers the semantic structure, regarding both global class alignment and local sample clustering, yielding improved accuracy. Across all domains, our method increases average accuracy by $3.51\%$ over the baseline. Note that current state-of-the-art JiGen [3] improves $1.86\%$ over its own baseline. In addition, we observe an improvement of $6.20\%$ when the unseen domain is *sketch*, which has a distinct style and requires more general knowledge about semantic concepts.

**Ablation analysis.** We conduct an extensive study using PACS benchmark to investigate two key points: 1) the contribution of each component to our method's performance, 2) how the semantic feature space is influenced by our proposed meta losses. First, we test all possible combinations of including the key components: episodic meta-learning simulating domain shift, global class alignment loss and local sample clustering loss. Accuracies averaged over three runs when using different combinations are given in Table 3. For example, first row corresponds to the DeepAll baseline with standard training by aggregating all source data. The fifth row is directly adding the $\mathcal{L}_{\text{global}}, \mathcal{L}_{\text{local}}$ losses on top of $\mathcal{L}_{\text{task}}$ with standard optimization scheme, i.e., without splitting $\mathcal{D}$ to meta-train and meta-test domains. From the ablation study, we observe that each component plays its own role in a complementary way. Specifically, the proposed losses that encourage semantic structure in feature space yield improvement over DeepAll, as well as over pure episodic training (the second row) that corresponds to our implementation of MLDG thus enabling straightforward comparison. By further leveraging the gradient-based update paradigm, performance is further improved across all settings.

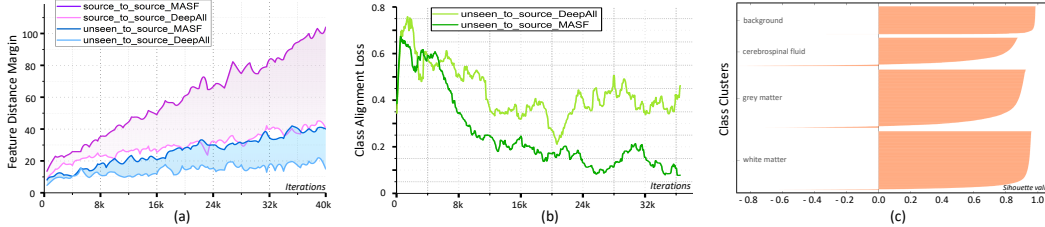

Figure 3: Analysis of the learning procedure: (a) margin of feature distance between sample negative pairs (with different classes) and positive pairs (with the same class), (b) class relationships alignment loss between unseen target domain and source domain, (c) Silhouette plot of the embeddings from meta metric-learning. Detailed analysis is in Section 4.2 for (a-b) and Section 4.3 for (c).

Table 3: Ablation study on key components of our method with the PACS dataset (accuracy, %).

| Episodic | $\mathcal{L}_{\text{global}}$ | $\mathcal{L}_{\text{local}}$ | Art | Cartoon | Photo | Sketch | Average |
|---|---|---|---|---|---|---|---|
| - | - | - | 67.60±0.21 | 68.87±0.22 | 89.20±0.24 | 61.13±0.30 | 71.70 |
| ✓ | - | - | 69.19±0.10 | 70.66±0.37 | 90.36±0.18 | 59.89±0.26 | 72.52 |
| - | ✓ | - | 69.43±0.29 | 70.22±0.21 | 90.64±0.15 | 60.11±0.17 | 72.60 |
| - | - | ✓ | 69.50±0.15 | 70.25±0.13 | 90.12±0.12 | 63.02±0.12 | 73.22 |
| - | ✓ | ✓ | 69.48±0.20 | 71.15±0.16 | 90.16±0.15 | 64.73±0.34 | 73.88 |
| ✓ | ✓ | - | 69.94±0.15 | 72.16±0.28 | 90.10±0.12 | 63.54±0.13 | 73.93 |
| ✓ | - | ✓ | 69.50±0.20 | 71.44±0.34 | 90.16±0.15 | 64.97±0.28 | 74.02 |
| ✓ | ✓ | ✓ | 70.35±0.33 | 72.46±0.19 | 90.68±0.12 | 67.33±0.12 | 75.21 |

Table 4: PACS results with deep residual network architectures (accuracy, %).

| Source | Target | ResNet-18 | | ResNet-50 | |
|---|---|---|---|---|---|
| | | DeepAll | MASF (ours) | DeepAll | MASF (ours) |
| C,P,S | Art-painting | 77.38 ± 0.15 | 80.29 ± 0.18 | 81.41 ± 0.16 | 82.89 ± 0.16 |
| A,P,S | Cartoon | 75.65 ± 0.11 | 77.17 ± 0.08 | 78.61 ± 0.17 | 80.49 ± 0.21 |
| A,C,S | Photo | 94.25 ± 0.09 | 94.99 ± 0.09 | 94.83 ± 0.06 | 95.01 ± 0.10 |
| A,C,P | Sketch | 69.64 ± 0.25 | 71.69 ± 0.22 | 69.69 ± 0.11 | 72.29 ± 0.15 |

We utilize t-SNE [52] to analzye the feature space learned with our proposed model and the DeepAll baseline (cf. Fig. 2). It appears that our MASF model yields a better separation of classes. We also note that the *sketch* domain is further apart from *art painting* and *cartoon*, although all three are source domains in this experiment, possibly explained by the unique characteristics of sketches. In Figure 3 (a), we plot the difference of feature distances between samples of negative pairs and positive pairs, i.e., $\mathbb{E}[\|\mathbf{z}_a - \mathbf{z}_n\|_2 - \|\mathbf{z}_a - \mathbf{z}_p\|_2]$. For the two magenta lines, respectively for MASF and DeepAll, sample pairs are drawn from different training source domains. We see that both distance margins naturally increase as training progresses. The shaded area highlights that MASF yeilds a higher distance margin between classes compared to DeepAll, indicating that sample clusters are better separated with MASF. Similarly, for the two blue lines, sample pairs are from the unseen target domain and a source domain (randomly selected at each iteration). As expected, the margin is not as large as between training domains, yet our method still presents a notably bigger margin than the baseline. In Figure 3 (b), we plot $\ell_{\text{global}}$ quantifying differences of average class posteriors between unseen target domain and a source domain during training. We observe that the semantic inter-class relationships, conveying general knowledge about a recognition task, would not naturally converge and generalize to the unseen domain without explicit guidance.

**Deeper architectures.** In the interest of providing stronger baseline results, we perform additional preliminary experiments using more up-to-date deep residual architectures [17] with ResNet-18 and ResNet-50. Table 4 shows strong and consistent improvements of MASF over the DeepAll baseline in all PACS splits for both network architectures. This suggests our proposed algorithm is also beneficial for domain generalization with deeper feature extractors.

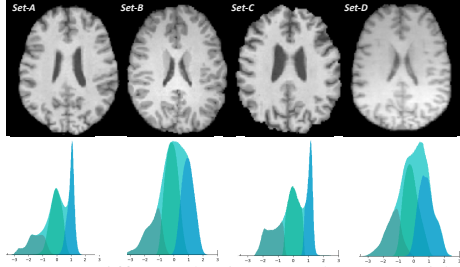

Figure 4: Different brain MRI datasets with example images and intensity histograms.

Table 5: Evaluation of brain tissue segmentation (Dice coefficient, %) in different settings: *columns 1–4:* train model on single source domain, test on all domains; *columns 5–6:* train on three source domains, test on remaining domain.

| Train<br>Test | Set-A | Set-B | Set-C | Set-D | DeepAll | MASF |
|---|---|---|---|---|---|---|
| Set-A | 90.62 | 88.91 | 88.81 | 85.03 | 89.09 | 89.82 |
| Set-B | 85.03 | 94.22 | 81.38 | 88.31 | 90.41 | 91.71 |
| Set-C | 93.14 | 92.80 | 95.40 | 88.68 | 94.30 | 94.50 |
| Set-D | 76.32 | 88.39 | 73.50 | 94.29 | 88.62 | 89.51 |

## 4.3 Tissue Segmentation in Multi-site Brain MRI

We evaluate our method on a real-world medical imaging task of brain tissue segmentation in T1-weighted MRI. Data was acquired from four clinical centers (denoted as Set-A/B/C/D). Domain shift occurs due to differences in scanners, acquisition protocols and many other factors, posing severe limitations for translating learning-based methods to clinical practice [13]. Figure 4 shows example images and intensity histograms. We adapt MASF for the segmentation of four classes: background, grey matter (GM), white matter (WM), cerebrospinal fluid (CSF). We employ a U-Net [38], commonly used for this task. For $\mathcal{L}_{\text{global}}$, the $\bar{\mathbf{z}}_c^{(k)}$ is computed by averaging over all pixels of a class. Our metric-embedding has two layers of $1 \times 1$ convolutions, with contrastive loss for $\mathcal{L}_{\text{local}}$. We randomly split each domain to $80\%$ for training and $20\%$ for testing in experimental settings.

**Results.** For easier comparison, we average the evaluated Dice scores achieved for the three foreground classes (GM/WM/CSF) and report it in Table 5. Although hard to notice visually from the gray-scale images, the domain shift from data distribution degrades segmentation significantly by up to $10\%$. DeepAll is a strong baseline, yet our model-agnostic learning scheme provides consistent improvement over naively aggregating data from multiple sources, especially when generalizing to a new clinical site with relatively poorer imaging quality (i.e., Set-D). Figure 3 (c) is the Silhouette plot [39] of the embeddings from $M_\phi$, demonstrating that the samples within the same class cluster are tightly grouped, as well as clearly separated from those of other classes.

## 5 Conclusions

We have presented promising results for a new approach to domain generalization of predictive models by incorporating global and local constraints for learning semantic feature spaces. The better generalization capability is demonstrated by new state-of-the-art results on popular benchmarks and a dense classification task (i.e., semantic segmentation) for medical images. The proposed loss functions are generally orthogonal to other algorithms, and evaluating the benefit of their integration is an appealing future direction. Our learning procedure could also be interesting to explore in the context of generative models, which may greatly benefit from semantic guidance when learning low-dimensional data representations from multiple sources.

## Acknowledgements

This project has received funding from the European Research Council (ERC) under the European Union's Horizon 2020 research and innovation programme (grant No 757173, project MIRA, ERC-2017-STG) and is supported by an EPSRC Impact Acceleration Award (EP/R511547/1). DCC is also partly supported by CAPES, Ministry of Education, Brazil (BEX 1500/2015-05).

## Footnotes

[1]For image segmentation, $F_\psi$ extracts *feature maps* and the task network $T_\theta$ is applied pixel-wise.

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
