[Reviews · NeurIPS 2019]

Reviewer 1



My main concern is that the problem of domain generalization is not very well-defined. It is necessary to explicitly specify the underlying assumption of the proposed method. "Directly generalize to target domains with unknown statistics" as mentioned in the abstract would be impossible if the target domain is very different from the training domains. In comparison, MAML has a task distribution from which the tasks (including target) are sampled. If MASF uses the same assumption, then the problem is not really domain generalization but meta-learning from multiple tasks as in MAML. The experiments focus on learning from multiple domains then applied the model to *only one* target domain, which is, in fact, more related to the multi-source to single-target adaptation than domain generalization. For example, the papers below [a,b,c] are theoretically justified and it would make sense to discuss and compare to them. [a] Mansour, Y., Mohri, M., and Rostamizadeh, A. (2009). Domain adaptation with multiple sources. In Advances in Neural Information Processing Systems, pages 1041-1048. [b] Zhao, H., Zhang, S., Wu, G., Moura, J. M., Costeira, J. P., and Gordon, G. J. (2018). Adversarial multiple source domain adaptation. In Advances in Neural Information Processing Systems, pages 8559-8570. [c] Hoffman, J., Mohri, M., and Zhang, N. (2018). Algorithms and theory for multiple-source adaptation. In Advances in Neural Information Processing Systems, pages 8246-8256. The necessity of global class alignment (Sec.3.2) is not very convincing: certain design choices are not clearly explained. It is not clear why the soft labels need to be computed based on average class features. We can alternatively compute the average soft labels given the data from one particular class directly (i.e., averaging over the final predictions instead of features). It is also not clear why symmetrized KL divergence is used instead of Jensen-Shannon divergence. Any comments or explanations on these alternatives would be helpful. What is the "linear-sized random subset" in L188? Experiments - The VLCS, PACS and MRI results in Table 1, 2 and 4 have no error bars. Are these results from one single run? Besides, it is not clear what the error bar in Table 3 means. Is it standard deviation, standard error or something else? Are these results statistically significant? - Why clipped gradient is needed? This indicates the proposed algorithm is not very stable or easy to train. - How is the margin parameter \xi selected in the experiments? And what criterion is the selection based on? - Compared to Table 1, some alternative methods are not included in Table 2. Why? Minors: - L95, samples are "drawn from" a dataset -> "drawn to form"? - L219, the highest ==================================== Update after rebuttal ==================================== The rebuttal resolves some of my concerns about the underlying assumption and design choices. It is essential to provide explicit assumptions about the proposed method. It also indicates that several hyperparameters are chosen heuristically. Without proper selection strategy or seeing sensitivity analysis about these hyperparameters, it is difficult to tell whether the improvements are due to better objective function or extensive hyperparameter-tuning. Overall, this is a borderline paper.

Reviewer 2



Originality/Significance: Although the proposed losses are established in other application areas/problem settings, and although their use in this application is kind of “obvious in retrospect”, these losses were not previously used in DG setting together with meta-learning. So it's good to highlight their efficacy in this problem setting. But otherwise the novelty is limited as the same meta-learning pipeline proposed for few-shot in MAML [10], and extended to domain generalisation in MLDG [23] is used. Engineering Issues: (i) Many hyper parameters are introduced (e.g. Algo 1). Tuning these is not straightforward in DG problems. (ii) L_global and L_local both trigger second order gradients, which makes training slow. (iii) L_global seems to compute all pairs of available meta-train and meta-test, which is slow and scales badly with number of domains. Empirical: (i) Recent DG papers [1] used ResNet rather than out of date AlexNet. If this paper is accepted, ResNet experiments should be included to avoid making the paper be of out-of-date relevance before its even published. Minor: - Some recent methods like [28] missing from comparison table. - L.219 highes -> highest 
 Overall assessment: It’s quite a “vision style” paper. Not really a fundamental machine learning development. However the motivation, explanation, ablation, and numerical performance are all quite good, and the real medical application is an icing on the cake. So it could be acceptable for NIPS. ----- Update. I have read the author feedback and other reviews. Besides the somewhat vision style, I don't see any real flaws. The updated addition of ResNet experiments will benefit the longevity and relevance of the paper. Hopefully the authors will also share code so others can build on it.

Reviewer 3



[After the author response] Thank you for reporting the performance of global alignment on JiGen as a baseline, which shows consistent performance boost. I intended to check whether the proposed method only works with DeepAll or not. I conjecture that the proposed method works well with the other methods not only JiGen. Also, it would be better if the performances of local loss are also reported in a future version of the paper. ============================================================== I’m positive to this paper because I believe that episodic training is an important topic for learning scheme. The proposed algorithm consists of three parts; Episodic training, global alignment, and local objective. The main contribution of this paper is that global alignment and local objective are adopted in the episodic training procedure. While each component seems to be closely related to the other methods, the combination of all components as the name of episodic training is plausible to solve domain generalization problem. While the proposed algorithm is validated on several benchmarks, there exists only a single baseline algorithm. I think that validating the proposed components, such as global alignment and local objective, on the other baseline algorithms strengthens the proposed algorithm.

[Author Response · NeurIPS 2019]

1 We thank the reviewers for the positive comments, highlighting that our *'motivation, explanation, ablation and*
2 *numerical performance are all quite good'*, with *'consistent improvement over the other algorithms'* on two benchmarks
3 comparing ten different methods, and an *'icing-on-the-cake real medical application'*. Both R2 and R3 point out the
4 contribution of combining global and local feature alignments with episodic training for domain generalization (DG).

5 R1's main concern regarding the DG problem setting is extensively clarified. We ran additional experiments regarding
6 ResNet-18/50 for R2, and JiGen as baseline for R3 (please see the table at the end). This rebuttal also clarifies all other
7 minor questions. We will add all these in the final version to further strengthen our contribution.

**Response to Reviewer #1**

**Problem setting: (i)** DG considers how to learn a model for a *single task* from a number of source domains and test it
on unseen domains, in contrast with MAML's assumption of training on a variety of learning tasks for solving new
tasks. **(ii)** We also clarify that DG is different from domain adaptation (DA), as DG assumes *no data is available* from
the target domain during training (L22–25). **(iii)** In DG, source and target domains correspond to joint distributions
$P_k(\mathbf{x}, y)$ and $P_*(\mathbf{x}, y)$ defined over input and label spaces $\mathcal{X} \times \mathcal{Y}$. It assumes there exist domain-invariant patterns
(i.e. *semantic features*) in the marginals $P_k(\mathbf{x})$ and $P_*(\mathbf{x})$, which can be extracted to learn an estimate of $P(y \mid \mathbf{x})$ that
performs well across seen and unseen domains. **(iv)** Thanks for pointing out the theoretical papers on multi-source to
single-target adaptation; we will revise the Sec. 2 accordingly. Our DG definition and experimental setting follow the
wide literature [1, 12, 22–25, 27, 31, 32] on this topic, but we agree a more theoretical discussion would be beneficial.

**Design choices: (i)** The class-specific average feature $\bar{\mathbf{z}}_c^{(k)}$ is considered as a compact semantic 'concept' of each class.
Computing soft labels from the features, rather than averaging final predictions, reflects our goal of explicit regularization
in feature space. **(ii)** We found no major theoretical reason to prefer Jensen–Shannon (JS) over symmetrized KL (a.k.a.
Jeffreys divergence) in our context. In preliminary experiments, we did try JS but obtained worse empirical results.
**(iii)** The 'linear-sized random subset of pairs' (L188) means that we can obtain an efficient unbiased $O(N)$ estimator of
the loss by e.g. shuffling and iterating over $(2i - 1, 2i)$, $i = 1, \ldots, \lfloor N/2 \rfloor$, rather than enumerating $O(N^2)$ pairs.

**Experiments: (i)** All results reported for our method and baselines are the average over 3 runs. Error bars in Table 3
are standard deviation. We will add error bars and statistical significance to Tables 1, 2, and 4. **(ii)** We clip the gradients
to prevent them from exploding, because our inner meta-update needs to be implemented with plain gradient descent
(not using an off-the-shelf optimizer). This follows the practice of MAML. **(iii)** We chose the margin $\xi$ heuristically,
based on preliminary observations of the distances within and between the clusters of class features. **(iv)** Tables 1 and 2
have different columns because not all of those papers reported results on both benchmarks.

**Response to Reviewer #2**

**Engineering issues: (i)** We had no difficulty in setting the hyperparameters (e.g. learning rates and loss coefficients).
Our heuristic choices worked well and other trials did not show much change. **(ii)** Computing second-order gradients
does not excessively slow down training—in MAML [10] (basis of our meta-learning scheme), it is roughly 33% slower
than a first-order approximation. **(iii)** Our $\mathcal{L}_{\mathrm{global}}$ can scale to numerous domains, by randomly sampling subsets of
meta-train and meta-test domains at each iteration, similarly to how MAML uses mini-batches of tasks. As our datasets
have only few domains, we used all of them ($|\mathcal{D}_{\mathrm{tr}}|=2$ and $|\mathcal{D}_{\mathrm{te}}|=1$, with one hold-out test domain).

**ResNet backbone:** Thank you for the suggestion. We now ran experiments with ResNet-18/50 on the PACS benchmark.
Our initial results shown in the table (mean $\pm$ std. dev. over 3 runs) are very promising, where our MASF consistently
improves over DeepAll baseline. We will add more systematic ResNet experiments in the final version.

**Response to Reviewer #3**

**Local loss:** Note that we employ *either* contrastive *or* triplet loss for local alignment, but not both simultaneously.
While contrastive loss has cheaper computational cost, it enforces much tighter constraints than triplet loss. As we
argue in L179–182, contrastive loss is a good choice for complex tasks with mild domain shift (e.g. medical image
segmentation), and triplet loss is adopted when domains are radically different (e.g. PACS benchmark).

**Additional baseline:** We reproduced the results of JiGen [3] using their released code, and present here preliminary
results of using JiGen as the baseline with our proposed $\mathcal{L}_{\mathrm{global}}$ (mean $\pm$ std. dev. over 3 runs). We find that our global
alignment is indeed complementary to JiGen's task-agnostic loss. Thanks for the inspiring suggestion. We will further
study the generic efficacy of our global and local semantic feature alignments in future work.

| | ResNet-18 | | ResNet-50 | | JiGen as baseline (with AlexNet) | | |
| Domain | DeepAll | MASF (ours) | DeepAll | MASF (ours) | JiGen [3] | Reproduced | $+\mathcal{L}_{\mathrm{global}}$ |
|---|---|---|---|---|---|---|---|
| Art-painting | $77.38 \pm 0.15$ | $80.29 \pm 0.18$ | $81.41 \pm 0.16$ | $82.89 \pm 0.16$ | 67.63 | $67.60 \pm 0.06$ | $68.36 \pm 0.10$ |
| Cartoon | $75.65 \pm 0.11$ | $77.17 \pm 0.08$ | $78.61 \pm 0.17$ | $80.49 \pm 0.21$ | 71.71 | $71.82 \pm 0.17$ | $71.91 \pm 0.11$ |
| Photo | $94.25 \pm 0.09$ | $94.99 \pm 0.09$ | $94.83 \pm 0.06$ | $95.01 \pm 0.10$ | 89.00 | $89.66 \pm 0.12$ | $89.80 \pm 0.09$ |
| Sketch | $69.64 \pm 0.25$ | $71.69 \pm 0.22$ | $69.69 \pm 0.11$ | $72.29 \pm 0.15$ | 65.18 | $65.52 \pm 0.15$ | $66.73 \pm 0.15$ |

[Meta-Review · NeurIPS 2019]

This is an interesting application work of domain generalization to the medial domain. In the rebuttal, the authors successfully address most concerns raised by reviewers on the definition of the problem setting, and some experimental setting issues. The authors are encouraged to revise the paper based on their rebuttal as well as the comments given by reviewers.